## RESEARCH ARTICLE

# IFT139 regulates Hedgehog signaling and cilia structure through ciliary protein localization

Khatija Nishat, Zachary Klug, Jannatul Faimma Mia, Sara M. Stump and Yulu Cherry Liu*

## ABSTRACT

The primary cilium, a microtubule-based membrane protrusion, is essential for eukaryotic development and health. Import and export of proteins in and out of the primary cilium relies on intraflagellar transport protein complexes (IFT) IFT-B and IFT-A, in conjunction with their respective motor proteins. Here, using mouse fibroblast cells, we investigated the function of IFT139 (Thm1, *TTC21B*) in Hedgehog signaling, cilia structure, and ciliary protein localization, as well as the effect of the P209L ciliopathy mutation on cell proliferation and Hedgehog signaling. In cells without IFT139, Ptch1 retains normal localization, Smo and Gli accumulate in the distal tips of cilia with or without pathway activation, while SuFu fails to accumulate in cilia upon pathway activation. We also found that Arl13b abnormally accumulates at the distal tips of cilia, but acetylated tubulin does not. Lastly, the ciliopathy mutation P209L impairs cell proliferation and Hedgehog transcriptional response, mimicking a loss of function in IFT139. Our work highlights the multifaceted roles IFT139 have on distinct ciliary proteins, and its importance in ciliopathies.

KEY WORDS: Cilia, Flagella, Signaling, Hedgehog

## INTRODUCTION

The primary cilium is a microtubule-based cell membrane protrusion found in almost all eukaryotic cells. One non-motile cilium is found per cell. It is essential for development where it functions as a signaling hub. The primary cilium originates from the basal body derived from the mother centriole. Its axoneme is made up of nine microtubule doublets enclosed in ciliary membrane, an extension of the plasma membrane (Reiter and Leroux, 2017; Mill et al., 2023; Hilgendorf et al., 2024). Within the cilium, microtubules orient with their plus-ends pointing in the direction of the distal tip. Two large protein complexes, intraflagellar transport complexes (IFT) A and B, work with their respective motor proteins to move proteins in and out of the cilium. Kinesin (KIF3) facilitates anterograde (toward the distal tip) transport in association with the IFT-B complex, whereas dynein mediates retrograde transport (toward the basal body) with the IFT-A complex. In addition to retrograde trafficking of cargo proteins, IFT-A complex is important for anterograde movement of membrane proteins into the cilium using specific adaptor proteins (Reiter and Leroux, 2017; Mill et al., 2023; Reddy Palicharla and

Mukhopadhyay, 2024). Studies have shown that disruption of KIF3 or IFT-B complex eliminates the primary cilium and results in diverse and severe developmental and cell signaling defects (Huangfu and Anderson, 2005; Liu et al., 2005). In contrast, mutations in dynein or IFT-A complexes result in stumpy cilia with proteins abnormally accumulating at the distal tips (Huangfu and Anderson, 2005; May et al., 2005).

The IFT-A complex consists of six functionally non-redundant proteins divided into two subcomplexes. IFT139 [alternative names: TTC21b (TPR-containing Hedgehog modulator 1), Thm1, NPHP12, SRTD4], IFT121 (WDR35), and IFT43 make up the peripheral complex, whereas the IFT122, IFT144 (WDR19), and IFT140 are in the core complex (Hirano et al., 2017; Behal et al., 2012; Mukhopadhyay et al., 2010; Jordan and Pigino, 2021). Mutations in individual IFT-A proteins have overlapping yet distinct phenotypes, and it has been speculated that these IFT-A proteins have cell specific roles or specific cargo recognition roles (Wang et al., 2020; Jordan and Pigino, 2021). The specificity of cargo protein binding is only beginning to be resolved. For example, the tubulins that make up the microtubule axoneme are transported into the primary cilium by IFT-B proteins, IFT74 and IFT81 (Bhogaraju et al., 2013; Jordan and Pigino, 2021), whereas IFT-A protein, IFT121 has been shown to be important for the selection and transportation of membrane proteins (Fu et al., 2016; Quidwai et al., 2021).

Cargo selectivity for IFT139 of the IFT-A complex is still largely unknown. IFT139 is a 130 amino acid protein with multiple tetratricopeptide repeat (TRP) domains. Early mouse genetics studies using ENU induced mutagenesis identified *Aln* mutant (IFT139[Aln]) with neural tube defects like those seen in other IFT or Hedgehog (Hh) mutations (Tran et al., 2008; Stottmann et al., 2009; Snedeker et al., 2017). In mice, global deletion of the IFT139 resulted in perinatal lethality, polydactyly, skeletal and forebrain defects, phenotypes also seen in loss of function of other IFT-A genes (Braun and Hildebrandt, 2017). Specific to IFT139, perinatal loss of IFT139 can lead to obesity in adulthood (Jacobs et al., 2016, 2020), and global deletion of IFT139 in juvenile or adult mice can lead to cystic kidneys that are reminiscent of autosomal dominant polycystic kidney disease (Tran et al., 2014; Wang et al., 2022). In humans, mutations in IFT139 have been identified in about 5% of patients with first order ciliopathies such as Jeune asphyxiating thoracic dystrophy, Joubert syndrome, and nephronophthisis. Patients exhibit polydactyly, abnormal thoracic rib cage development and kidney defects (Huynh Cong et al., 2014; Braun and Hildebrandt, 2017; Davis et al., 2011; Mill et al., 2023). Biochemically, less is known about the specific role of IFT139 in cargo recognition and protein trafficking. From cell-based assays, IFT139 binds strongly with its subcomplex partners IFT121 and IFT43, and loss of IFT139 from cells leads to shortened or stumpy cilia and abnormal accumulation of ciliary proteins (Hirano et al., 2017; Tran et al., 2008, 2014). In this paper, using mouse fibroblast cells with or without IFT139, we examined in detail the localization of Hh

Department of Biology, Hood College, Frederick, Maryland, 21701, USA.

*Author for correspondence (liuyc@hood.edu)

Y.C., 0000-0003-4861-6915

Biology Open

signaling proteins, acetylated tubulin and Arl13b, and studied the effect of ciliopathy disease mutation P209L in cell proliferation and Hh response.

## RESULTS

Primary cilia are essential for proper Hh signal transduction (Huangfu and Anderson, 2005; Kopinke et al., 2021; Ingham, 2022; Hilgendorf et al., 2024). During pathway activation, Hh ligand binds to its receptor, Patched-1 (Ptch1), resulting in the exit of Ptch1 from the primary cilium (Rohatgi et al., 2007), and the accumulation of Smoothened (Smo) in the cilium (Corbit et al., 2005). Subsequently, several other Hh pathway components, SuFu, Kif7, Gli2, and Gli3 all accumulate at the distal tip of the cilium, which then results in the nuclear localization of Gli2 and Gli3 and transcriptional activation of the pathway (Haycraft et al., 2005; Cheung et al., 2009; Endoh-Yamagami et al., 2009; Liem et al., 2009; Kim et al., 2009; Tukachinsky et al., 2010). To study the role of IFT139 in cilia and Hh signaling, we examined Hh proteins' localization in wild-type cells (WT) (litter mate control) mouse fibroblasts (MEF), and cells from IFT139[Aln] mice where there is no IFT139 protein expression (Tran et al., 2008) (Fig. S1A). Using stable expression of Ptch1GFP in these MEF cells and Arl13b as a cilia marker, we examined the ciliary localization of Ptch1 in WT and IFT139[Aln] cells. Ptch1GFP exited from the cilium upon pathway activation with Sonic Hedgehog (Shh) conditioned media treatment (Fig. 1A, left, quantified in B), and the same was observed in IFT139[Aln] cells (Fig. 1A, right, quantified in B). Patched-2 (Ptch2) is a paralog of Ptch1 (Zaphiropoulos et al., 1999) with redundant functions in skin and skeletal development (Adolphe et al., 2014; Zhulyn et al., 2015). To examine Ptch2 ciliary localization, we made a cell line stably expressing Ptch2GFP and found that Ptch2 exited from the cilium in WT and in IFT139[Aln] cells upon pathway activation (Fig. S1B,C), but the difference was not statistically significant in IFT139[Aln] cells. Overall, our data suggests that loss of IFT139 does not impact Ptch1 cilia localization in cells.

In WT cells, Smo and Gli accumulate in the cilium upon pathway activation via treatment with Shh-conditioned media or with the Smo agonist (SAG) (Chen et al., 2002) (Fig. 1C, left, quantified in D and E). In agreement with previous reports (Tran et al., 2008; Hirano et al., 2017; Wang et al., 2021), cells without IFT139 showed abnormal accumulation of Smo and Gli at the distal tips of cilia in the absence of Hh pathway stimulation, and that treatment with Shh or SAG cannot stimulate Smo to WT level (Fig. 1C, right, quantified in D and E). To see if the abnormal accumulation of Smo in the cilium can be reversed pharmacologically, we treated cells with the Smo antagonist, SANT, which inhibits Smo cilia localization in WT cells (Fig. S1D) (Chen et al., 2002). However, SANT treatment did not reverse the abnormal Smo or Gli accumulation observed in IFT139[Aln] cells (Fig. S1D,E). This indicates that in cells without IFT139, the abnormal localization of Smo and Gli are not due to mis-regulated activity of Smo protein itself.

Gli ciliary localization is known to be negatively regulated by PKA. Activation of PKA by forskolin (Fsk) can inhibit Gli ciliary accumulation (Fig. S1E) (Tukachinsky et al., 2010; Zeng et al., 2010). To see if the abnormal accumulation of Gli in the cilium of IFT139[Aln] cells can be reversed, we treated these cells with Fsk. Relative to DMSO control, Fsk treatment did not inhibit the abnormal accumulation of Gli (Fig. S1E). This suggests that Fsk-based activation of PKA cannot reverse the defect seen in cells without IFT139.

The suppressor of fused (SuFu), a negative regulator of the pathway, is also known to localize to the primary cilium upon pathway activation (Tukachinsky et al., 2010) (Fig. 1F, left, quantified in G). Unlike Gli, SuFu never accumulates in the cilium of IFT139[Aln] cells, even with pathway activation by Shh or SAG (Fig. 1F, right, quantified in G). This observation suggests that IFT139 might be controlling the timely localization of Gli and Smo in response to pathway activation at a step that is after Ptch1 exit but before Gli-SuFu complex forms at the base of the cilium upon pathway activation (Fig. 1I).

At the transcriptional level, upon pathway activation, the target gene *Gli1* is turned on and results in a large increase of *mGli1* mRNA when assayed (Tukachinsky et al., 2010; Liu et al., 2014; Liu et al., 2022). Here, using RT-qPCR to quantify pathway activation at the transcriptional level, we found that in IFT139[Aln] cells there is poor pathway activation in the presence of Shh or SAG, as compared to WT cells (Fig. 1H) (Wang et al., 2020), suggesting that the abnormally accumulated Smo and Gli in IFT139[Aln] cells are not contributing to pathway activation. In titrating Shh-conditioned media, we found that IFT139[Aln] cells require higher Shh dosages than WT cells to initiate Gli1's transcriptional response *in vitro* (Fig. S1F). This might help to explain why IFT139 mutations have varying degrees of severity in neural tube defects (ventral versus dorsal) where there is a gradient of Hh morphogen. This might also help to explain why mutation of IFT139 can partially rescue the defects in Gli2 or Smo mutations (Tran et al., 2008; Stottmann et al., 2009).

Previous works on IFT139 mutant or null cells have reported a bulging (widened distal tip) and stumpy (shortened) defect in the cilium structure (Tran et al., 2008, 2014; Davis et al., 2011; Hirano et al., 2017; Wang et al., 2020). To examine the structural defects in IFT139[Aln] cells in detail, we performed immunofluorescence on WT and IFT139[Aln] MEF cells for acetylated tubulin (AcTub, magenta) and Arl13b (cyan). In WT cells, acetylated tubulin and Arl13b showed overlapping localization along the entire axoneme (Fig. 2A, left). However, in IFT139[Aln] cells, Arl13b accumulated in a bulge at the distal tip of cilium, whereas acetylated tubulin accumulated in the proximal or basal half of the cilium axoneme (Fig. 2A, right), as marked by the basal body protein centrin (Fig. S2A). Quantification of fluorescent intensities for Arl13b and acetylated tubulin showed significant reductions in intensities of both proteins in cilia axonemes overall (Fig. 2B,C). 3D reconstructions of the cilium from Arl13b image stacks (Fig. 2A, bottom panel) further illustrate the bulging phenotype, as measured by the width (red color) of the cilium at the distal tip. Consistent with previous studies, there is also a decrease in average cilia length in IFT139[Aln] cells compared to WT cells (Fig. 2D) (Davis et al., 2011; Tran et al., 2014; Wang et al., 2020). However, in contrast to the previous report (Wang et al., 2020), we did not observe a significant difference in ciliogenesis (percentage of cells with cilia) between WT and IFT139[Aln] cells (Fig. 2E). To further quantify the structural defects observed in IFT139[Aln] cells, we compared torsion (cilium axoneme twisting out of a plane) (Fig. 2F) and curvature (cilium axoneme curving within a plane) (Fig. 2G), denoted as the inverse of the radius (1/μm). We found that there were no significant differences between the two cell types, suggesting that despite the bulging at the distal tip of the cilium, the axonemal structure appears to be similar between WT and IFT139[Aln] cells.

It is possible that the stumpy cilia phenotype is caused by premature disassembly of the organelle. Primary cilia are disassembled through activation of aurora kinase A (Pugacheva et al., 2007). Studies have shown that inhibiting aurora kinases with small molecule inhibitors can stop cilia disassembly (Korobeynikov et al., 2017; Kiseleva et al., 2019). Here, using aurora kinase A inhibitor (MLN8054) (Sells et al., 2015) and aurora kinase B inhibitor (AZD1152) (Yang et al., 2007), we tested if the shortened cilia length observed in IFT139[Aln] cells is due to early or premature cilia disassembly. We found that treatment

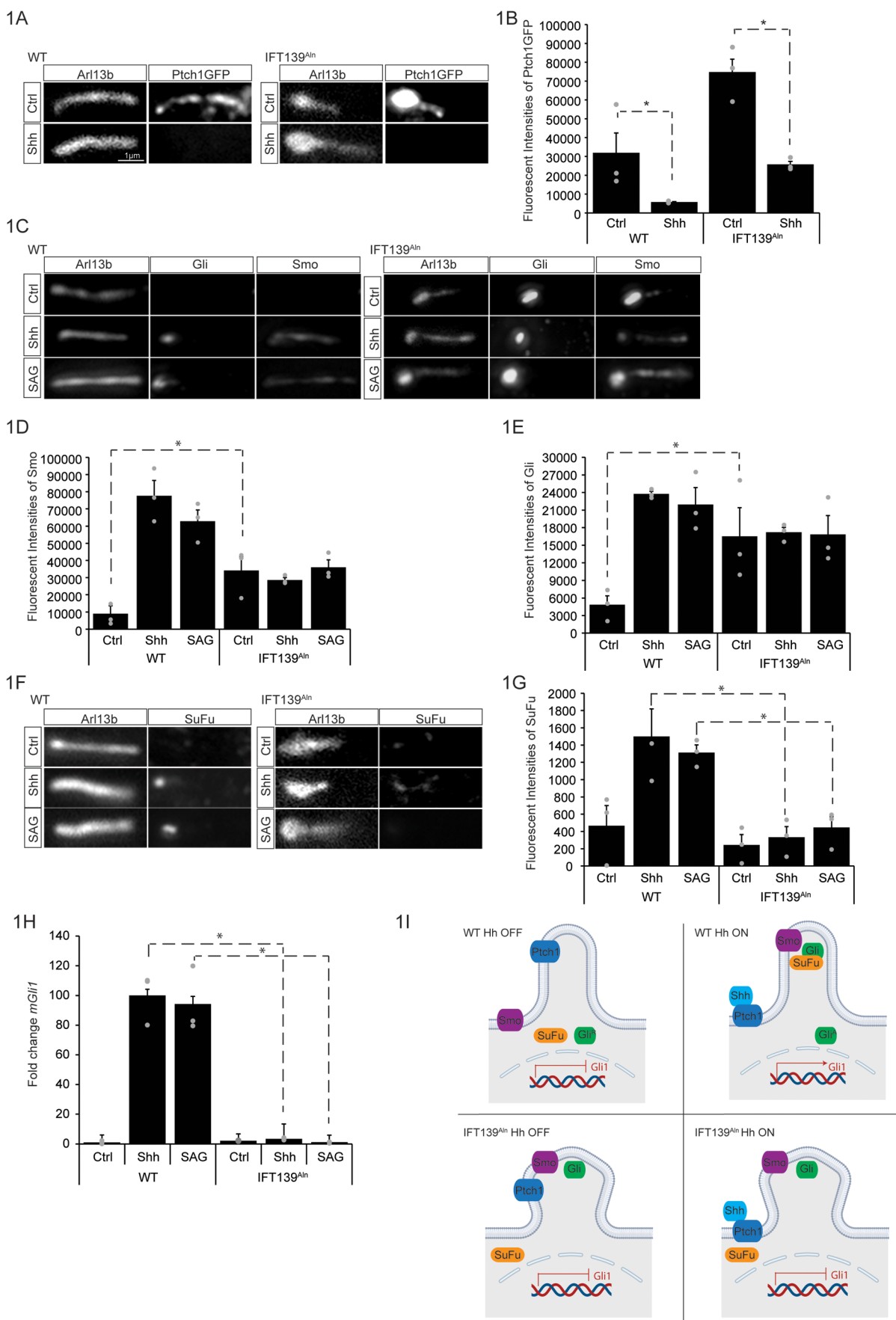

**Fig. 1.** See next page for legend.

**Fig. 1. IFT139 is necessary for the ciliary localization of Hh proteins and pathway activation.** (A) Images of WT or IFT139[Aln] cells stably expressing Ptch1GFP following treatment with control media (ctrl) or Shh-conditioned media (Shh). Ciliary intensity of cilia was detected by staining for endogenous Arl13b. Scale bar: 1 µm. (B) Quantification of the experiment in A. Bars represent average ciliary intensity of Ptch1GFP and error bars show SD. 100 cilia were measured per condition per experiment for three independent experiments (grey dots). * denotes statistical significance, $P<0.05$ by one-way ANOVA. (C) Images of WT or IFT139[Aln] cells stained for endogenous Gli and Smo following treatment with control media (ctrl), Shh-conditioned media (Shh), or SAG (1 µM). Cilia were detected by staining for endogenous Arl13B. Scale bar: 1 µm. (D,E) Quantification of the experiment in C. Bars represent average ciliary intensity of Smo (D) or Gli (E) and error bars show SD. 100 cilia were measured per condition per experiment for three independent experiments (grey dots). * denotes statistical significance, $P<0.05$ by one-way ANOVA. (F) Images of WT or IFT139[Aln] cells stained for endogenous SuFu following treatment with control media (ctrl), Shh-conditioned media (Shh), or SAG (1 µM). Cilia were detected by staining for endogenous Arl13b. Scale bar: 1 µm. (G) Quantification of the experiment in F. Bars represent average SuFu ciliary intensity and error bars show SD. 100 cilia were measured per condition per experiment for three independent experiments (grey dots). * denotes statistical significance, $P<0.05$ by one-way ANOVA. (H) WT or IFT139[Aln] MEFs were incubated with control (ctrl) media, Shh-condition media (Shh), or SAG (1 µM), and Hh signaling was measured by qRT-PCR for *mGli1*. Bars show average fold-change for three replicates (grey dots) and error bars show SD. Data were normalized from 0% (ctrl) to 100% activation of the Hh pathway by Shh. * denotes statistical significance, $P<0.05$ by one-way ANOVA. (I) Schematic representations of the results in Fig. 1. Top panel: Hh pathway in WT cells. Gli[A], active form of Gli that turns on transcription of target gene; Gli[R], repressive form of Gli that inhibits transcription of target gene. Bottom panel: Hh pathway changes in IFT139[Aln] cells.

with the two kinase inhibitors did not change cilia length (Fig. S2B) or acetylated tubulin intensity (Fig. S2D) in IFT139[Aln] cells compared to WT cells. There was a small but significant increase in Arl13b fluorescent intensity with aurora kinase A inhibitor (MLN8054) treatment compared to control treatment, but not with aurora kinase B inhibitor (AZD1152) (Fig. S2C). These data suggest that the decrease in cilia length might not be due to cilia disassembly. Lithium chloride (LiCl) has been shown experimentally to increase cilia length (Miyoshi et al., 2009; Nakakura et al., 2014; Thompson et al., 2016), possibly through modulation of membrane via actin filament regulator Arp2/3 complex (Bigge et al., 2023). However, LiCl treatment had no significant effect on IF139[Aln] cells' cilia length or fluorescent intensities of Arl13b (Fig. S2C) or acetylated tubulin (Fig. S2D), supporting the conclusion that LiCl treatment can increase cilia length via membrane deformations and not through changes in ciliary protein levels (Bigge et al., 2023). Altogether, results from these experiments suggest that the defects observed in IFT139[Aln] cells: short cilia length, distal accumulation of Arl13b, and proximal distribution of acetylated tubulin cannot be rescued through small molecules that modulate cilia disassembly or cilia length.

Multiple mutations in IFT139 have been found in various ciliopathy patient groups (Halbritter et al., 2013; Braun and Hildebrandt, 2017). One mutation, P209L, in which proline 209 is conserved between human and mouse, has been found in multiple studies. Patients with this mutation suffer from a range of symptoms; renal fibrosis and liver cirrhosis (Gambino et al., 2021), kidney glomerular and tubulointerstitial damages (Huynh Cong et al., 2014), arterial hypertension and tubuloglomerular kidney disease (Olinger et al., 2022), and nephronophthisis-associated ciliopathies (Halbritter et al., 2013; Otto et al., 2011; Davis et al., 2011). To better study the molecular basis of this point mutation, we generated a knock-out of IFT139 in mouse fibroblast C3H10T/12 cells using CRISPR-Cas9

(Ran et al., 2013) (Fig. S3A,B). Using lentiviral transduction, we then stably expressed WT human IFT139 or the human P209L mutant in WT or knock-out (KO) C3H10T/12 cells (Fig. S3B,C) and compared the proliferation of these four cell lines in culture via a cell counting assay validated in Fig. S3D. Compared to WT cells, KO cells grew slower, and by day 3 had more than 60% reduction in total cell number (Fig. 3A). Expression of WT hIFT139 in IFT139KO cells (green) partially rescued the growth defect in KO cells (Fig. 3B). However, expression of hIFT139P209L in IFT139 KO cells (light blue) failed to rescue the knockout (Fig. 3B), suggesting that the P209L mutation mimics a loss-of-function mutation. To further support this, we introduced P209L mutant into WT cells and found that this mutant exhibits a dominant negative growth phenotype. Expression of hIFT139P209L mutant in WT mouse fibroblast cells (green) impaired cell growth (Fig. 3C) to a similar level as seen in KO cells (Fig. 3A). Lastly, to examine Hh signaling in these cells, we performed RT-qPCR to quantify pathway activation at the transcriptional level. Similar to what we found in Fig. 1H with MEF cells, C3H10T1/2 IFT139KO cells showed poor pathway activation in the presence of SAG as compared to WT cells (Fig. 3D). Expression of hIFT139 partially rescued the *mGli1* expression defect in IFT139KO cells treated with SAG, whereas hIFT139P209L did not. This finding is analogous to what we observed with cell proliferation assay (Fig. 3B). Together, these data suggest that the P209L mutant is a loss-of-function mutation in cell-based assay, and that when expressed with WT protein, it behaves as a dominant negative, perturbing the function of the WT copy.

## DISCUSSION

The Hh signaling pathway is reliant on the precise localization of proteins into and out of the primary cilium (Kopinke et al., 2021; Hilgendorf et al., 2024). Here we investigated the role of IFT139 in the ciliary localization of several known Hh pathway components. We found that while Ptch1 exhibited normal ciliary movements, Smo, Gli, and SuFu did not in cells without IFT139. Specifically, Smo and Gli accumulated abnormally in the cilium in pathway off state, and SuFu failed to accumulate in the cilium at all (Fig. 1I). These ciliary localization defects lead to the abolition of overall Hh signaling at the transcriptional level. Our data suggests that IFT139 is needed for the proper localization of Gli-Sufu complex into the cilium upon pathway activation, and without it, the pathway fails to turn on transcriptionally. Given the importance of Hh signaling in skeletal development it is possible that the skeletal defects seen in patients with IFT139 mutations are due to dysregulation of the Hh pathway (Kopinke et al., 2021; Hilgendorf et al., 2024).

Previous works have found that cells without IFT139 have stumpy and bulging cilia (Tran et al., 2008, 2014). We show here that in IFT139[Aln] cells, Arl13b accumulates abnormally at the distal tips of cilia, which become bulbous. In contrast, acetylated tubulin localized mostly in the proximal half of the cilium axoneme. These deformations are likely not caused by improper alignment of the axoneme, as its torsion and curvature are unaffected by loss of IFT139. They are likely to result from protein localization defects. It is possible that Arl13b is a cargo specific to IFT139 (Fu et al., 2016) and that without IFT139, Arl13b retrograde transport is impaired leading to its accumulation at the distal tip of the cilium. Another possibility is that IFT139 is important for the proper binding of IFT-A complex to dynein, and therefore necessary for retrograde trafficking (Nakayama and Katoh, 2018). As such, in the absence of IFT139, there would be poor removal of Arl13b from the cilium. Further experiments would be necessary to validate either of these models.

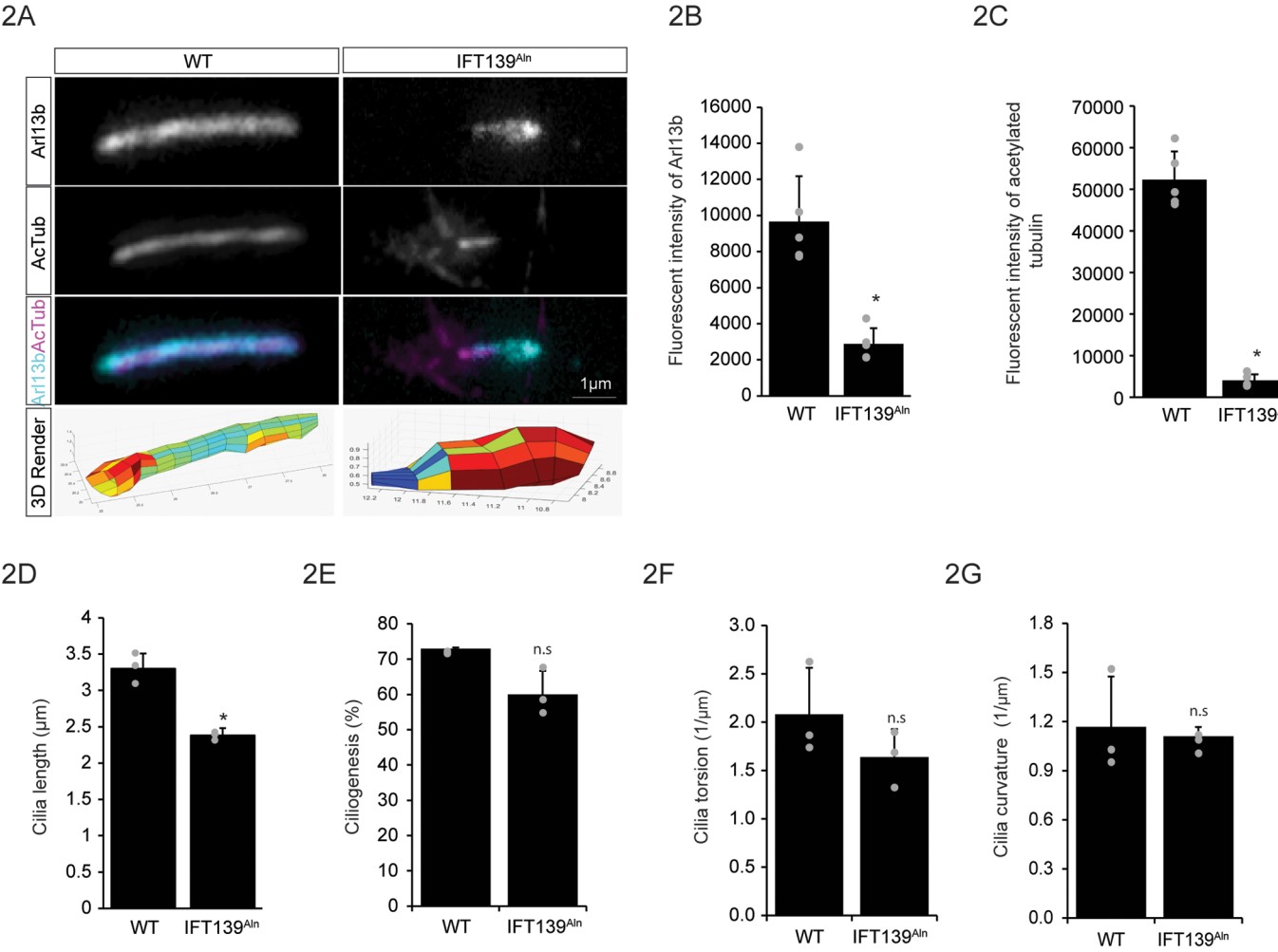

**Fig. 2. Loss of IFT139 results in stumpy cilia and changes in Arl13b and acetylated tubulin cilia localization.** (A) Immunofluorescence images of the primary cilium in WT or IFT139<sup>Aln</sup> cells. Cilia were stained for Arl13b (cyan) and acetylated tubulin (AcTub) (magenta). Scale bar: 1 μm. 3D reconstructions of the cilium from Arl13b image stacks are shown at the bottom panel. The color scale is an indication of the width of the distal tip of the cilium: red indicates wide, blue indicates narrow. Axes are arbitrary units. (B) Quantification of the experiment in A. Bars represent average ciliary intensity of Arl13b and error bars show s.d. 100 cilia were measured per cell type per experiment for five independent experiments (grey dots). * denotes statistical significance, $P<0.05$ by paired $t$-test. (C) Quantification of the experiment in A. Bars represent average ciliary intensity of acetylated tubulin and error bars show s.d. 100 cilia were measured per cell type per experiment for five independent experiments (grey dots). * denotes statistical significance, $P<0.05$ by paired $t$-test. (D) Quantification of cilia length (μm) in WT or IFT139<sup>Aln</sup> MEF cells. Bars represent average cilia length (Arl13b staining) and error bars show s.d. 150 cilia were measured per cell type per experiment for three independent experiments (grey dots). * denotes statistical significance, $P<0.05$ by paired $t$-test. (E) Quantification of ciliogenesis (%) in WT or IFT139<sup>Aln</sup> MEF cells. Bars represent average percentage of total cilia number divided by total number of cells and error bars show s.d. 150 cilia were measured per cell type per experiment for three independent experiments (grey dots). n.s. denotes not statistically significant, $P>0.05$ by paired $t$-test. (F) Quantification of cilia torsion in WT or IFT139<sup>Aln</sup> MEF cells. Bars represent average torsion or cilium twisting out of a plane (1/μm, inverse of the radius) and error bars show s.d. 150 cilia were measured per cell type per experiment for three independent experiments (grey dots). n.s denotes not statistically significant, $P>0.05$ by paired $t$-test. (G) Quantification of cilia curvature in WT or IFT139<sup>Aln</sup> MEF cells. Bars represent average curvature, cilium axoneme curving within a plane (1/μm, inverse of the radius) and error bars show s.d. 150 cilia were measured per cell type per experiment for three independent experiments (grey dots). n.s denotes not statistically significant, $P>0.05$ by paired $t$-test.

Multiple IFT139 mutations have been found in ciliopathy patients. The P209L mutation was found in multiple studies. It has been suggested that P209L might function as a hypomorph in podocytes (Huynh Cong et al., 2014). However, in our cell-based assays, we found P209L to mimic a loss-of-function mutation and act as a dominant negative in the presence of the WT protein. AlphaFold predictions of the structural changes resulting from the mutation of the phylogenetically conserved proline 209 to leucine found that this change likely causes steric clashes in the third TPR domain, a type of domain commonly used for protein–protein interactions (Olinger et al., 2022). Recent structural studies of IFT-A complex showed that IFT139's TPR domains are crucial for contact

with IFT74 of the IFT-B complex, and that this contact is important for the polymerization of IFT-A proteins onto the anterograde train at the base of the cilium (Brown et al., 2015; Jordan et al., 2018; Meleppattu et al., 2022; Ma et al., 2023; Lacey et al., 2023). Since P209L mutation is in the third TPR domain, it is possible that this mutation might hinder the assembly of IFT-A onto IFT-B via IFT74, which would inhibit proper localization of IFT-A itself into the cilium and subsequently affect its function.

One hypothesis that we did not test is that IFT-A regulates the selection of specific cargo proteins outside of the cilium, and is needed to transport the cargo protein across the transition zone, a protein based interface between the basal body and the cilium

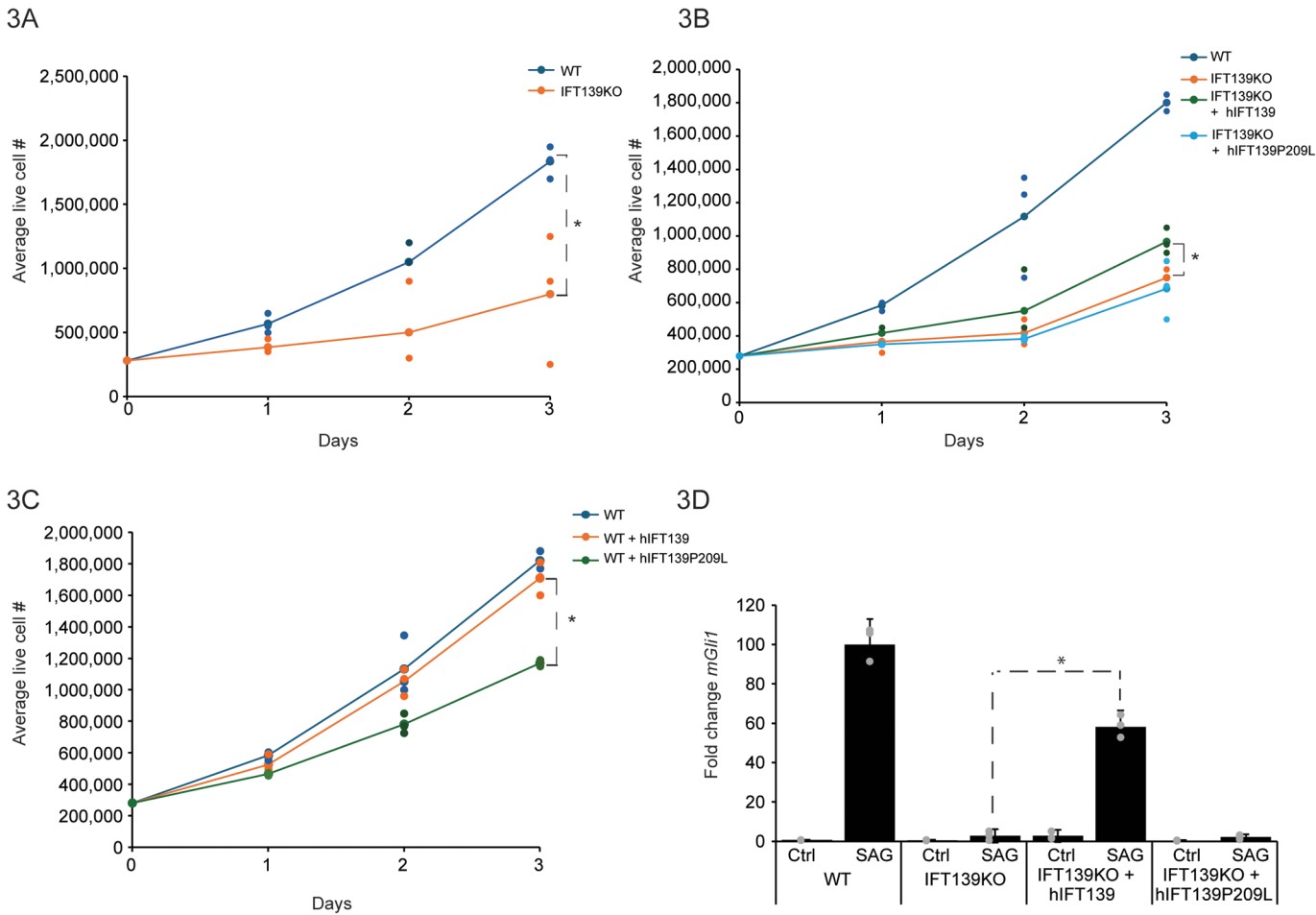

**Fig. 3. Expression of IFT139 P209L mutation impaired cell proliferation and Hh signaling response.** (A) Cell proliferation was measured as the average live cell number. WT (dark blue) or IFT139KO (orange) C3H10T1/2 cells were counted for three consecutive days. The lines represent average live cell numbers with the three replicates represented as colored dots. * denotes statistical significance, $P<0.05$ by one-way ANOVA. (B) Cell proliferation was measured as the average live cell number. WT (dark blue), IFT139KO (orange), IFT39KO expressing hIFT139 (green), or IFT139KO expressing hIFT139P209L (light blue) C3H10T1/2 cells were counted for three consecutive days. The lines represent average live cell numbers with the three replicates represented as colored dots. * denotes statistical significance, $P<0.05$ between IFT139KO (orange) and IFT39KO expressing hIFT139 (green) by one-way ANOVA. (C) Cell proliferation was measured as the average live cell number. WT (dark blue), WT expressing hIFT139 (orange), or WT expressing hIFT139P209L (green) C3H10T1/2 cells were counted for three consecutive days. The lines represent average live cell numbers with the three replicates represented as colored dots. * denotes statistical significance, $P<0.05$ by one-way ANOVA. (D) WT or IFT139KO C3H10T1/2 cells expressing hIFT139 or hIFT139P209L were treated with control (ctrl) or SAG (1 μM), and Hh signaling was measured by qRT-PCR for *mGli1*. Bars show average fold-change for three replicates (grey dots) and error bars show s.d. Data were normalized from 0% (ctrl) to 100% activation of the Hh pathway by SAG. * denotes statistical significance, $P<0.05$ by one-way ANOVA.

that functions like a selective diffusion barrier between the cilium and the cell body (Scheidel and Blacque, 2018; Jordan and Pigino, 2021). It would be interesting to further examine transition zone protein localization in P209L or IFT139 null cells. Another aspect that we did not test is the roles of Thm2 on Hh signaling and cilia formation. Thm2, or *TTC21A* is a paralog of IFT139 that also localizes to the cilium and may function together with IFT139 in Hh signaling (Allard et al., 2021; Bumann et al., 2022).

## MATERIALS AND METHODS
### Cell lines, cell culture, and media preparation
Human embryonic kidney cells (HEK293T), MEFs, and mouse fibroblasts (C3H10T1/2) were cultured in Dulbecco's Modified Eagle's Medium (DMEM) supplemented with 10% (v/v) fetal bovine serum (FBS), 1% penicillin/streptomycin, and 1% GlutaMax (ThermoFisher). Phosphate-buffered saline (PBS) was prepared by diluting 10× PBS stock (ThermoFisher) to 1× using sterile distilled water. Stable cell lines were

generated by infection with lentiviruses expressing genes of interest. Briefly, genes of interest were subcloned into the third-generation lentiviral vector pHAGE (Mostoslavsky et al., 2006), which was used to produce lentiviruses in HEK293T cells, as previously described (Wierbowski et al., 2020). Lentiviruses were mixed with 1 μg/ml hexadimethrine bromide (polybrene) (Sigma) and were used to infect the desired target cells. 48 h post-infection, stably transduced cells were isolated by selection with blasticidin (10 μg/ml) (ThermoFisher) or hygromycin (100 μg/ml) (ThermoFisher).

### Cell counting and proliferation assay
C3H10T/12 mouse fibroblast cells were seeded on 60 mm plates at a density of 280,000 cells per plate on day zero and then counted every 24 h for 3 days. For counting, cells were washed with 1 ml 1× PBS, then lifted with trypsin. Trypsin was then neutralized with 4 ml of complete media, and 1 ml of the cell suspension was transferred to a tube. Cells (200 μl) were counted using Vial-Cassette (ChemMetec) with built-in dye on the NucleoCounter (NC200) (ChemoMetec). Viable, live cell numbers were recorded. Every sample was counted three times. Cells were then put back on to the plates to continue growth.

## Gene editing by CRISPR-Cas9

Guide RNA (gRNA) sequence was designed to target murine *IFT139/Ttc21b* (http://chopchop.cbu.uib.no). Synthetic oligonucleotides (IDT) containing gRNA sequences were annealed and cloned into the pX459 vector (Ran et al., 2013). Parental cells were transfected (Lipofectamine 2000) with the gRNA-expressing plasmids and were transiently selected with puromycin (2 µg/ml) (ThermoFisher). The cells were then plated and grown clonally. Genomic DNA was extracted from individual clones, and the gRNA target loci was PCR amplified, Sanger sequenced and subjected to TIDE analysis (https://tide.nki.nl/) (Brinkman and van Steensel, 2019). The KO cell line was further validated via RT-qPCR. gRNA and primers used are in Table S1.

## Antibodies

For immunofluorescence, primary and secondary antibodies were used at 1 µg/ml in TBST supplemented with 5% bovine serum albumin (BSA). Primary antibodies for immunofluorescence were: rabbit anti-GFP (Rockland, #600-401-215), mouse anti-acetylated tubulin (Sigma, #T7451), chicken anti-Arl13b (made by Salic lab, details are described in Petrov et al., 2020), rabbit anti-Gli (made by Salic lab, details are described in Tukachinsky et al., 2010), goat anti-Smo (made by Salic lab, details are described in Tukachinsky et al., 2010), and mouse anti-SuFu (made by Salic lab, details are described in Tukachinsky et al., 2010). Fluorophore-conjugated secondary antibodies were as follows: donkey anti-chicken IgY–Alexa Fluor 647 (Jackson ImmunoResearch), donkey anti-goat IgG–Alexa Fluor 594 (Jackson ImmunoResearch), donkey anti-rabbit IgG–Alexa Fluor 488 (ThermoFisher), donkey anti-mouse IgG Alex Fluor 594 (ThermoFisher), and donkey anti-mouse IgG Alex Fluor 488 (ThermoFisher).

## Shh-conditioned media

Shh-conditioned media was produced as previously described (Nedelcu et al., 2013). Briefly, an expression construct encoding the first 197 amino acids of human Shh cloned in the pCS2 vector was transiently transfected into HEK293T cells, using polyethyleneimine (PEI). The next day, the medium was replaced with DMEM, and the cells were incubated for 48 h. The conditioned media was collected, centrifuged to remove cellular debris, and then used in signaling assays. Shh media was used at 1 in 10 dilution, unless otherwise indicated. Control media from untransfected cells was collected in parallel.

## Chemicals

Forskolin (no. S2249), SAG (no. S7779), MLN8054 (no. S1100) and AZD1152 (no. S1147) were from Selleckchem. LiCl was from ThermoFisher. SANT-1 (14933) was from Cayman Chemical.

## DNA constructs

Constructs were generated by PCR and were subcloned into the lentiviral pHAGE vector, driven by a human CMV promoter. The pHAGE constructs were used to produce lentiviruses, for generating stable cell lines. The construct for expressing mouse Ptch1 or Ptch2 tagged with eGFP at the C-terminus was described previously (Tukachinsky et al., 2016). Human IFT139/Ttc21b (NM_024753.5) was cloned from the Harvard Medical School cDNA collection, primers used are listed in Table S1.

## Immunofluorescence microscopy

MEF cells were plated on 12-mm diameter gelatin-coated round glass coverslips, in 24-well plates, at a density of $1 \times 10^5$ cells per well. Following overnight incubation, the complete medium was replaced with serum-free DMEM to induce ciliogenesis. After 24 h, the cells were treated with the indicated factors in serum-free DMEM, for another 24 h. The cells were then fixed in PBS with 3.7% formaldehyde, for 20 min at room temperature. Following permeabilization with PBST, endogenous Smo, endogenous Gli2/3, endogenous SuFu, or overexpressed Ptch1GFP, Ptch2GFP, or CentrinGFP (gift from A. Salic) was detected by immunofluorescence microscopy. Cells were co-stained for endogenous Arl13B, to detect primary cilia. The stained coverslips were mounted in PBS with 50% glycerol and were imaged on a Nikon TE2000E wide-field epifluorescence microscope, equipped with an OrcaER camera (Hamamatsu) and a 40×

PlanApo 0.45NA air objective (Nikon), as previously described (Nedelcu et al., 2013). For each condition, MetaMorph software (Molecular Devices) was used to acquire z-series consisting of five focal planes for at least 30 fields of view, for fluorescence channels corresponding to Arl13B, SuFu, AcTub, and Smo, Gli or GFP. The z series were used to generate maximum intensity projections, which were analyzed using custom image analysis scripts written in FIJI (NIH) and MATLAB (Mathworks) (Nedelcu et al., 2013). Briefly, cilia were segmented by local adaptive thresholding of Arl13B images and, for each cilium, the background-corrected fluorescence intensity for SuFu, Smo, Gli or eGFP fluorescence intensity was calculated. Data are presented as mean +/- SD across biological replicates. *$P<0.05$. Cilia length, torsion, and curvature were all measured in MATLAB based on segmented Arl13b. Torsion is measuring axoneme twisting out of a plane, whereas curvature is measuring axoneme curving within the plane, both are denoted as 1/µm, inverse of the radius. Both are calculated using MATLAB's Curvature-Torsion Defined Curve function based on Frenet's equations.

## Quantitative reverse transcription PCR (qRT-PCR)

Cells were plated in triplicate in six-well plates and, after reaching confluency, were serum-starved overnight. Afterwards, the cells were treated with the indicated factors in DMEM, for 24 h. Total RNA was isolated using TRIzol (ThermoFisher), after which the RNA was treated with DNase I (Promega), followed by a second round of TRIzol purification. The RNA was reverse transcribed with Luna SuperScript (NEB) and random hexamers. Target genes were measured with Power SYBR Green (ThermoFisher) as previously described (Liu et al., 2014). The primer sequences are listed in Table S1. The comparative $C_T$ method (Schmittgen and Livak, 2008) was used to compute expression of target gene relative to *cyclophilin*. For Hh pathway activation, data were normalized from 0% (ctrl) to 100% activation of the Hh pathway (saturating amount of the Shh). Data are presented as mean±s.e.m. across three biological replicates. *$P<0.05$.

## Data collection and statistical analysis

All experiments were performed in triplicates. Statistical significance was assessed using Excel's Student's paired *t*-test or GraphPad's one-way ANOVA, as indicated in figure caption. *, $P<0.05$; n.s, not statistically significant $P>0.05$.

## Acknowledgements

We thank Adrian Salic for sharing reagents, cell lines and antibodies. We thank Jonathan T. Eggenschwiler for IFT139[Aln] MEF cells. We thank Yanqing Xu for developing the MATLAB script for cilia segmentation and measurements. We thank CLIA Molecular Diagnostics Laboratory of the Frederick National Laboratory for help with Sanger sequencing. We thank Marco A. Catipovic for critical reading of the manuscript. We thank Meredith Yeager for help with sequence alignment and homology calculations.

## Competing interests

The authors declare no competing or financial interests.

## Author contributions

Conceptualization: Y.C.L., K.N., S.M.S.; Data curation: Y.C.L., K.N., Z.K., J.F.M., S.M.S.; Formal analysis: Y.C.L., K.N.; Funding acquisition: Y.C.L.; Investigation: Y.C.L.; Methodology: Y.C.L., Z.K., J.F.M., S.M.S.; Project administration: Y.C.L.; Resources: Y.C.L.; Software: Y.C.L.; Supervision: Y.C.L.; Validation: Y.C.L., S.M.S.; Visualization: Y.C.L.; Writing – original draft: Y.C.L., K.N.; Writing – review & editing: Y.C.L., K.N.

## Funding

This project is supported by Hood College's Summer Research Institute grant to Y.C.L. and S.M.S. and by U.S. Department of Education's FIPSE funding to the Biosciences Research and Education Center at Hood College, which supports Y.C.L. and K.N. Open Access funding provided by Hood College's FIPSE grant. Deposited in PMC for immediate release.

## Data and resource availability

All relevant data and details of resources can be found within the article and its supplementary information.

Biology Open

## First Person

This article has an associated First Person interview with the first author of the paper.

## Peer review history

The peer review history is available online at https://journals.biologists.com/bio/lookup/doi/10.1242/bio.062040.reviewer-comments.pdf

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
