## [Peer Review File · Biology Open]

IFT139 regulates Hedgehog signaling and cilia structure through ciliary protein localization

Khatija Nishat, Zachary Klug, Jannatul Faimma Mia, Sara M. Stump and Yulu Cherry Liu
DOI: 10.1242/bio.062040

Editor: Catherine L. Jackson

Review timeline

Original submission:	18 March 2025
Editorial decision:	28 April 2025
First revision received:	19 May 2025
Editorial decision:	20 August 2025
Second revision received:	26 August 2025
Accepted:	1 September 2025

Original submission

First decision letter

MS Title: IFT139 regulates Hedgehog signaling and cilia structure through ciliary protein localization.

Authors: Yulu Cherry Liu; Khatija Nishat; Zachary Klug; Jannatul Faimma Mia; Sara M Stump

Dear Dr Liu,

We have now reached a decision on the above manuscript.

To see the reviewers' reports and a copy of this decision letter, please go to:

As you will see from their reports, the reviewers raise a number of substantial criticisms that prevent me from accepting your paper for publication.

I am very sorry to give you such disappointing news, but we are currently under great pressure for space and it takes a very enthusiastic recommendation by the referees for a manuscript to be accepted.

I do hope you find the comments of the reviewers helpful in allowing you to revise the manuscript for submission elsewhere, and many thanks for sending your work.

Reviewer 1:

SUMMARY OF THE ADVANCE MADE IN THIS PAPER AND ITS POTENTIAL SIGNIFICANCE TO THE FIELD

In their manuscript, "IFT139 regulates Hedgehog signaling and cilia structure through ciliary protein localization", Nishat et al. investigate the impact of IFT139 mutations on the ciliary localization of several dynamic Hedgehog (Hh) pathway components, cilia length and shape, and cell proliferation using a cell-based approach. They show loss of IFT139 results decreased cilia length and altered ciliary protein composition and proliferative defects. Additionally, they demonstrate that IFT139

modulates Hh signaling at a step after Ptch ciliary exit but before the Gli-SuFu complex forms upon pathway activation. The experiments are properly conducted and controlled and the data support the conclusions. However, published work has already demonstrated these results so the authors' conclusions provide an incremental advance the current understanding of IFT139's function. Thus it is challenging to describe the results as cutting-edge science per the mission of JCS.

SUGGESTIONS TO AUTHORS

The authors make several observations and propose interesting ideas that seem worth pursuing with experiments. Some examples include: 1) the proposal that Arl13b may be an IFT139 cargo; 2) that the P209L mutation may interrupt IFT-A:IFT-B assembly; and 3) that P209L expression doesn't rescue the slow growth of IFT139KO cells suggesting there may be problems with cell cycle, ciliation rate, cilia length and/or Hh component trafficking / signaling.

MAJOR ISSUES

1. The IFT139KO MEFs used in Figures 1 and 2 are not clearly identified. Where did the cells come from? What is the mutation that makes them "KO"? Is Figure S1A qRT-PCR data?
2. In Figure 3, new IFT139KO mouse fibroblasts are generated through CRISPR-Cas9 mutation. Why switch cells? What mutation(s) occurred due to NHEJ repair in the cell line and what change would be seen at the protein level? How do these cells compare to those used in earlier experiments?
3. The CRISPR-generated IFT139KO cells used in Figure 3 are incompletely rescued by hIFT139 expression. Were multiple clones analyzed? What are some reasons for the poor rescue?
4. Data appears to be reused in Figure 3: the WT curve is present on all three graphs (same data?), and the IFT139KO curve is in 3A and 3B (same data?).
5. The methods used to measure cilia torsion and curvature are vague, "Custom image analysis scripts written in MATLAB". These need to be described with sufficient detail that others can repeat.

MINOR ISSUES:

1. Figure legends are difficult to understand. "As in (B), but...." Please make it easier on readers by just stating what is being shown, measured, or quantified.
2. "Fluorescent" is misspelled throughout the figures.
3. Figure S1C quantifies (B), not (A).
4. What cells are being counted in Figure S3D? The legend suggests there should be SD error bars.
5. Purchased primary antibodies in the methods section refer to previous publications instead of the company/catalog/RRID. Ex: "rabbit anti-GFP (Petrov et al 2020)" is better written as "rabbit anti-GFP (Rockland, Cat# 600-401-215; RRID: AB_828167)".
6. Check antibodies for accuracy:
 - i) I couldn't find "rabbit anti-Gli2/3 (Tukachinsky et al 2016)". In that reference's Materials section, there's a "goat anti-human Gli3 (R&D Systems)".
 - ii) The anti-SuFu antibody is listed as mouse, but no anti-mouse secondary antibody is listed.
 - iii) A Google search for "ThermoFisher 21-2700" returns no hits.
7. Citations in several places do not correspond to the associated statements. Ex: Ocbina and Anderson 2008, and Ocbina et al 2011 are inappropriately referenced twice in the Results section for Figure 1 (approx. line 51 on page 5, line 21 on page 6).
8. It is unclear why cell proliferation was chosen as a readout for the P209L mutation.

Reviewer 2:

SUMMARY OF THE ADVANCE MADE IN THIS PAPER AND ITS POTENTIAL SIGNIFICANCE TO THE FIELD

This work aims to evaluate the function of IFT139, a component of the IFT-A primary cilium (PC) transport machinery that transports cargo to the base of the PC. The authors evaluate localization of Hedgehog (Hh) pathway components in cells lacking IFT139 function (IFT139 KO) and/or expressing a mutant version of the protein that has been observed in ciliopathies (P209L). They also briefly examine ciliary architecture and find that cilia are short and stubby following P209L mutant expression. This result is consistent with previous reports.

Although the research described in the manuscript evaluates an important cell biological phenomenon, the study is too premature for publication. The authors demonstrate that Hh pathway components exhibit altered ciliary localization in cells lacking IFT139, but they don't test whether this is a specific effect or simply a consequence of altered ciliary function resulting from altered IFT-A function. Conclusions are made without results that directly support them. Many experiments lack essential controls and oftentimes relevant details about how the experiments were performed are lacking. The study does not appear to be hypothesis-driven, and many of the results provided are phenomenological. Because of this, the study comes across as lacking a clear objective or focus, necessitating a significant amount of additional work for the study to be considered for publication.

SUGGESTIONS TO AUTHORS

Key questions/suggestions for the authors:

1. Have the IFT139 KO cells been reported previously? How were they generated? I can't find any information in the methods or acknowledgements regarding the provenance of these cells.
2. How many cells were evaluated for the quantification of protein ciliary signal intensity? Please provide this information in the figure legends and please include an image showing a wider field of view so that the reader can appreciate ciliary localization of the various proteins across the cell population. The single PC images shown are fine to make the point, but don't give us an idea of what is happening across the population.
3. Figure 1C-D - why does loss of IFT139 cause more Smo to accumulate less at the tip of the PC in Shh stimulated cells than in non-stimulated cells (per 1C)?
4. The visual representation in 1C does not match what is shown for quantification in 1D. For example in the control cells, Smo + SAG signal looks much weaker than in the IFT139 KO cells, but according to the graph, there is less Smo in the PC. Please show zoom out images so the reader can determine whether the selected cilia are representative of the population.
5. Much of the data presented are negative data - i.e. no overt effect is observed from drug treatment - yet strong conclusions are made. For example, for treatment of IFT139KO cells with Aurora Kinase A or Aurora Kinase B inhibitor, no change in cilia length was observed - so the authors conclude the architecture change is "not due to cilia disassembly". Same theme for forskolin treatment and Hh Gli protein localization to the PC. Did you confirm that your drug is working as expected? Is there a positive control? Same question for LiCl treatment. Are there any other ways to modulate ciliary length genetically that you could combine with IFT139KO?
6. Figure 3 - Please show protein expression of WT vs. P209L IFT139 protein in the 10T1/2 cells. As presented, it is impossible to know if the defects described are due to a specific effect of the mutation or to the protein levels not being equivalent.
7. Near line 28 on page 9, the authors state that they looked at transport of several Hh pathway components - technically, they did not look at transport. They looked at ciliary localization of these proteins in cells that had compromised ciliary architecture. It's not clear that the effects on Shh

pathway components are specific or just a result of the altered ciliary function that occurs with IFT139 loss. Thus, the data do not justify the title.

8. The authors propose in the last sentence of the manuscript that P209L mutation might be hindering assembly of IFT-A onto IFT-B. There is no data provided to support this conclusion.

Minor comments:

1. A diagram of the various IFT complexes and subcomplexes would be helpful for the nonspecialist reader.
2. A description of how the PC regulates Shh signaling is needed in the introduction.
3. Figure 2: please indicate the number of cells/cilia analyzed for each experiment in the figure legends.

Reviewer 3:

The manuscript used MEF to characterize the cellular defects in cells that lack IFT139, a component of the highly conserved IFT-B complex. The authors mainly focused on the trafficking of Shh pathway proteins as well as Arl13b, a membrane protein specifically localized to primary cilia. In addition, the effect of P209L, a ciliopathy associated mutation, was examined in the context of cell proliferation. The conclusion is largely consistent with the IFT139 mouse knock-out phenotypes and provide additional insight into the roles for IFT-A complex in ciliogenesis and Hh signaling.

Specific comments

* Instead of saying in Method > Immunofluorescence microscopy "100-200 cilia were counted per condition.", information of number of cells or cilia used in each quantification are required,

* For Fig. S1A and S3B, clear labeling on bar graphs about which cell line is used in generating IFT139 KO might help minimize confusion while interpreting the results, i.e. mouse fibroblasts (MEF) for S1A and mouse fibroblast C3H10T1/2 for S3B.

* Figures 2A and S2A

Centrin signal not very specific.

S2A does not seem consistent with the argument made in the text,

"However, in IFT139KO cells, Arl13b accumulated in a bulge at the distal tip of cilium, whereas AcTub accumulated in the proximal half of the cilium"

How consistent were the cilia morphology reported in 2A observed?

* Is there any verification of protein level in IFT139 KO cell lines using western blot analysis?

* Page 7, Line 21, "...IFT139KO cells require higher Shh dosages than wild-type cells to initiate a transcriptional response (Figure S1F)." Can we infer the presence of proper transcription response merely based on Gli1 fold change level?

* Page 8, Line 50, "To better study the molecular basis of one such point mutation, P209L (Davis et al., 2011; Otto et al., 2011; Halbritter et al., 2013; Huynh Cong et al., 2014; Gambino et al., 2021; Olinger et al., 2022), ..." Please elaborate more on background for studying P209L mutation in particular, such as clinical significance of P209L, before introducing findings from experiments?

* "Using lentiviral transduction, we stably expressed wild-type human IFT139 or the P209L mutant in wild-type (WT) or knock-out (KO) cells"

Murine cells were used to conduct transduction experiment, is there a reason why human KO cells were not used?

How conserved are murine and human IFT139?

* "To further quantify the structural defects observed in IFT139KO cells, we compared torsion (axoneme twist)"

For these morphological analysis, higher resolution imaging may be useful, whether it is better resolved IF images or SEM.

How were torsion and curvature measured, would it be possible to give further details in addition to stating Matlab scripts used.

Lastly, can the authors speculate whether the peripheral IFT-A vs core IFT-A play distinct roles in controlling ciliogenesis and Hh signaling?

First revision

Author response to reviewers' comments

Reviewer 1: SUMMARY OF THE ADVANCE MADE IN THIS PAPER AND ITS

POTENTIAL SIGNIFICANCE TO THE FIELD

In their manuscript, "IFT139 regulates Hedgehog signaling and cilia structure through ciliary protein localization", Nishat et al. investigate the impact of IFT139 mutations on the ciliary localization of several dynamic Hedgehog (Hh) pathway components, cilia length and shape, and cell proliferation using a cell-based approach. They show loss of IFT139 results decreased cilia length and altered ciliary protein composition and proliferative defects. Additionally, they demonstrate that IFT139 modulates Hh signaling at a step after Ptch ciliary exit but before the Gli-SuFu complex forms upon pathway activation. The experiments are properly conducted and controlled and the data support the conclusions. However, published work has already demonstrated these results so the authors' conclusions provide an incremental advance the current understanding of IFT139's function. Thus it is challenging to describe the results as cutting-edge science per the mission of JCS.

SUGGESTIONS TO AUTHORS

The authors make several observations and propose interesting ideas that seem worth pursuing with experiments. Some examples include: 1) the proposal that Arl13b may be an IFT139 cargo; 2) that the P209L mutation may interrupt IFT-A:IFT-B assembly; and 3) that P209L expression doesn't rescue the slow growth of IFT139KO cells suggesting there may be problems with cell cycle, ciliation rate, cilia length and/or Hh component trafficking/signaling.

MAJOR ISSUES

1. The IFT139KO MEFs used in Figures 1 and 2 are not clearly identified. Where did the cells come from? What is the mutation that makes them "KO"? Is Figure S1A qRT-PCR data?

- We apologize for the confusion, IFT139KO should be clearly identified as MEF cells taken from *Aln* mutant mice from Tran et al 2008. These cells have no expression of IFT139 protein as noted in Tran et al 2008. We re-labeled our data as IFT139 Δ ln.

To avoid confusion, in this document, we will refer to this as IFT139 null cells in our responses to reviewer comments.

- Figure S1A is RT-qPCR data. We clarified this in the text and in figure legend.

2. In Figure 3, new IFT139KO mouse fibroblasts are generated through CRISPR-Cas9 mutation. Why switch cells? What mutation(s) occurred due to NHEJ repair in the cell line and what change would be seen at the protein level? How do these cells compare to those used in earlier experiments?

- We switched cell line because we could not rescue the MEF IFT139 Δ ln cells. Lentiviral expressions did not work with those cell lines after repeated attempts.

- In the C3H10T1/2 cells, the mutation is a single nucleotide insertion in exon 2. We did not perform NGS due to cost, instead we performed TIDE analysis (Brinkman and van Steensel, 2019). RT-qPCR in S3B shows that there is little mRNA expression for IFT139 in KO cells.

- We did not perform Western blot to assess protein level because we tried with one IFT139 antibody, but the antibody was not specific, and we do not have the funding to buy any more antibodies.

- C3H10T1/2 cells have been reported to behave like MEFs in forming cilia and are Hedgehog signaling responsive (Liu et al 2014). C3H10T1/2 IFT139KO cells also showed impaired Hedgehog signaling response via RT-qPCR (Figure 3D) similar to what we observed with MEF IFT139Aln cells.

3. The CRISPR-generated IFT139KO cells used in Figure 3 are incompletely rescued by hIFT139 expression. Were multiple clones analyzed? What are some reasons for the poor rescue?

- The rescue cell lines were polyclonal. The incomplete rescue could be due to the polyclonal nature of the cells.

4. Data appears to be reused in Figure 3: the WT curve is present on all three graphs (same data?), and the IFT139KO curve is in 3A and 3B (same data?).

- Data is not reused. All data points on Figure 3 were independently generated with three replicates. We added individual data points.

5. The methods used to measure cilia torsion and curvature are vague, "Custom image analysis scripts written in MATLAB". These need to be described with sufficient detail that others can repeat.

- We added in the equation used to generate cilia torsion and curvature in the main text and in the figure legend, and better defined both terms. We also added more details in materials and methods.

MINOR ISSUES:

1. Figure legends are difficult to understand. "As in (B), but...." Please make it easier on readers by just stating what is being shown, measured, or quantified.

- We apologize for the confusion. We truncated figure legends to reduce word count for the short report submission in Journal of Cell Science. We changed back to the original detailed version of the figure legends.

2. "Fluorescent" is misspelled throughout the figures.

- We corrected this.

3. Figure S1C quantifies (B), not (A).

- We corrected this.

4. What cells are being counted in Figure S3D? The legend suggests there should be SD error bars.

- Wild type C3H10T1/2 mouse fibroblast cells are counted in this experiment. We added individual data points for this experiment.

5. Purchased primary antibodies in the methods section refer to previous publications instead of the company/catalog/RRID. Ex: "rabbit anti-GFP (Petrov et al 2020)" is better written as "rabbit anti-GFP (Rockland, Cat# 600-401-215; RRID: AB_828167)".

- We apologize for the confusion. We corrected this. All antibodies are correctly referenced with either catalog # or citation.

6. Check antibodies for accuracy:

i) I couldn't find "rabbit anti-Gli2/3 (Tukachinsky et al 2016)". In that reference's Materials section, there's a "goat anti-human Gli3 (R&D Systems)".

ii) The anti-SuFu antibody is listed as mouse, but no anti-mouse secondary antibody is listed.

iii) A Google search for "ThermoFisher 21-2700" returns no hits.

- i) We corrected this. It is Tukachinsky et al 2010.

- ii) We corrected this. We added in the secondary mouse antibody.

- iii) We corrected this. It is Sigma #T7451.

7. Citations in several places do not correspond to the associated statements. Ex: Ocbina and Anderson 2008, and Ocbina et al 2011 are inappropriately referenced twice in the Results section for Figure 1 (approx. line 51 on page 5, line 21 on page 6).

- We corrected this. We removed these two citations.

8. It is unclear why cell proliferation was chosen as a readout for the P209L mutation.

- Cell proliferation was chosen as a readout because it was an affordable readout that we could conduct.

Reviewer 2: SUMMARY OF THE ADVANCE MADE IN THIS PAPER AND ITS POTENTIAL SIGNIFICANCE TO THE FIELD

This work aims to evaluate the function of IFT139, a component of the IFT-A primary cilium (PC) transport machinery that transports cargo to the base of the PC. The authors evaluate localization of Hedgehog (Hh) pathway components in cells lacking IFT139 function (IFT139 KO) and/or expressing a mutant version of the protein that has been observed in ciliopathies (P209L). They also briefly examine ciliary architecture and find that cilia are short and stubby following P209L mutant expression. This result is consistent with previous reports.

Although the research described in the manuscript evaluates an important cell biological phenomenon, the study is too premature for publication. The authors demonstrate that Hh pathway components exhibit altered ciliary localization in cells lacking IFT139, but they don't test whether this is a specific effect or simply a consequence of altered ciliary function resulting from altered IFT-A function. Conclusions are made without results that directly support them.

Many experiments lack essential controls and oftentimes relevant details about how the experiments were performed are lacking. The study does not appear to be hypothesis-driven, and many of the results provided are phenomenological. Because of this, the study comes across as lacking a clear objective or focus, necessitating a significant amount of additional work for the study to be considered for publication.

SUGGESTIONS TO AUTHORS

Key questions/suggestions for the authors:

1. Have the IFT139 KO cells been reported previously? How were they generated? I can't find any information in the methods or acknowledgements regarding the provenance of these cells.

- Yes, this is a previously reported cell line. We apologize for the confusion, IFT139KO should be clearly identified as MEF cells taken from *Aln* mutant mice from Tran et al 2008. These cells have no expression of the IFT139 protein. We have re-labeled our data as IFT139*Aln*

2. How many cells were evaluated for the quantification of protein ciliary signal intensity? Please provide this information in the figure legends and please include an image showing a wider field of view so that the reader can appreciate ciliary localization of the various proteins across the cell population. The single PC images shown are fine to make the point, but don't give us an idea of what is happening across the population.

- We added in the numbers of cell quantified per experiment.

- The quantification of the ciliary fluorescence signal across all cilia counted (100 per condition per experiment, for 3 independent repeats, total 300 cilia) best illustrates the distribution of the signal intensity. We also added individual average data points to the figure.

3. Figure 1C-D - why does loss of IFT139 cause more Smo to accumulate less at the tip of the PC in Shh stimulated cells than in non-stimulated cells (per 1C)?

- There is very little Smo at the tip of cilia in non-stimulated state, this is the lowest of all conditions. In IFT139 null cells (IFT139*Aln*), Smo accumulates in the cilia, specifically in the distal tip and cannot be further stimulated or increased by Shh or SAG. The Smo accumulating at the tip is not in the active state as if it was stimulated by Shh or SAG.

This data is similar to what has been published. We further clarified this in the main text.

4. The visual representation in 1C does not match what is shown for quantification in 1D. For example in the control cells, Smo + SAG signal looks much weaker than in the IFT139 KO cells, but according to the graph, there is less Smo in the PC. Please show zoom out images so the reader can determine whether the selected cilia are representative of the population.

- The quantification for 1D is for the entire cilium axoneme. The representative figures often look "off" because of the intense Smo accumulation at the distal tip of the cilium. Together, accounting

for distal tip and cilium axoneme, there is a reduction in Smo signal in IFT139 null cells in stimulated state.

5. Much of the data presented are negative data - i.e. no overt effect is observed from drug treatment - yet strong conclusions are made. For example, for treatment of IFT139KO cells with Aurora Kinase A or Aurora Kinase B inhibitor, no change in cilia length was observed - so the authors conclude the architecture change is "not due to cilia disassembly". Same theme for forskolin treatment and Hh Gli protein localization to the PC. Did you confirm that your drug is working as expected? Is there a positive control? Same question for LiCl treatment. Are there any other ways to modulate ciliary length genetically that you could combine with IFT139KO?

- The data referred to here for Aurora kinase inhibitors and LiCl are in Figure S2. Yes, we confirmed that the drugs are working. In this supplementary figure, we showed that drug treatments have a modest effect on cilia length in wild type cells (Figure S2B), and we did not observe any change to IFT139 null cells when treated with these drugs. We modified our wording in the text to ensure that we are not drawing strong conclusions here.

- The data referred to here for the forskolin experiment are in Figure S1. Yes, we confirmed that forskolin treatment works, as it reduced Gli level in wild type cells (Figure S1E). We also included SAG treatment to show that the assay is working. We modified our wording in the text to ensure that we are not drawing strong conclusions here.

- There are other ways to modulate cilia length genetically such as using IFT88KO or KIF3AKO cells which would abolish cilia. We do not have those cell lines.

6. Figure 3 - Please show protein expression of WT vs. P209L IFT139 protein in the 10T1/2 cells. As presented, it is impossible to know if the defects described are due to a specific effect of the mutation or to the protein levels not being equivalent.

- We did not perform Western blot to assess protein level because we tried with one

IFT139 antibody, but the antibody was not specific, and we do not have the funding to buy any more antibodies.

- From RT-qPCR, we can see there are expressions of hIFT139 and P209L mutant in mouse cells.

7. Near line 28 on page 9, the authors state that they looked at transport of several Hh pathway components - technically, they did not look at transport. They looked at ciliary localization of these proteins in cells that had compromised ciliary architecture. It's not clear that the effects on Shh pathway components are specific or just a result of the altered ciliary function that occurs with IFT139 loss. Thus, the data do not justify the title.

- We modified our language throughout the manuscript.

8. The authors propose in the last sentence of the manuscript that P209L mutation might be hindering assembly of IFT-A onto IFT-B. There is no data provided to support this conclusion.

- We modified our language to reflect that this is only speculative.

Minor comments:

1. A diagram of the various IFT complexes and subcomplexes would be helpful for the nonspecialist reader.

- We did not include a diagram for IFT-A complex because we only studied IFT139 in this paper. If the editors would like us to include a diagram we be happy to add one.

2. A description of how the PC regulates Shh signaling is needed in the introduction.

- We included this in the beginning section of the results section.

3. Figure 2: please indicate the number of cells/cilia analyzed for each experiment in the figure legends.

- We added this in.

Reviewer 3: The manuscript used MEF to characterize the cellular defects in cells that lack IFT139, a component of the highly conserved IFT-B complex. The authors mainly focused on the trafficking of Shh pathway proteins as well as Arl13b, a membrane protein specifically localized to primary cilia. In addition, the effect of P209L, a ciliopathy associated mutation, was examined in the

context of cell proliferation. The conclusion is largely consistent with the IFT139 mouse knock-out phenotypes and provide additional insight into the roles for IFT-A complex in ciliogenesis and Hh signaling.

Specific comments

* Instead of saying in Method > Immunofluorescence microscopy "100-200 cilia were counted per condition.", Information of number of cells or cilia used in each quantification are required,

- We corrected this.

* For Fig. S1A and S3B, clear labeling on bar graphs about which cell line is used in generating IFT139 KO might help minimize confusion while interpreting the results, i.e. mouse fibroblasts (MEF) for S1A and mouse fibroblast C3H10T1/2 for S3B.

- We corrected this. S1A is now labeled as IFT139AIn, whereas S3B is IFT139KO.

* Figures 2A and S2A

Centrin signal not very specific. S2A does not seem consistent with the argument made in the text, "However, in IFT139KO cells, Arl13b accumulated in a bulge at the distal tip of cilium, whereas AcTub accumulated in the proximal half of the cilium". How consistent were the cilia morphology reported in 2A observed?

- CentrinGFP is overexpressed in the cell line, and overexpression of Centrin is not healthy for cells (e.g. centrosomal duplication), fluorescent signals were not the best. We included this in the supplementary to illustrate orientation of the cilium, Centrin is at the base of the cilium.

- In Figure S2A right, IFT139 null cells, it shows that acetylated tubulin (red) occupies half of the cilium axoneme, whereas Arl13b (blue) occupies the distal half of the cilium. In comparison, in WT, Arl13b and acetylated tubulin overlaps, resulting in a pink color. This agrees with what we stated in the main text.

* Is there any verification of protein level in IFT139 KO cell lines using western blot analysis?

- Not in this manuscript, but previous work (Tran et al, 2008) showed by Western blot that there is no protein. We verified this via RT-qPCR due to cost.

* Page 7, Line 21, "...IFT139KO cells require higher Shh dosages than wild-type cells to initiate a transcriptional response (Figure S1F)." Can we infer the presence of proper transcription response merely based on Gli1 fold change level?

- We revised our language here to reflect that we only assessed Gli1 and not any other transcriptional factors.

* Page 8, Line 50, "To better study the molecular basis of one such point mutation, P209L (Davis et al., 2011; Otto et al., 2011; Halbritter et al., 2013; Huynh Cong et al., 2014; Gambino et al., 2021; Olinger et al., 2022), ..." Please elaborate more on background for studying P209L mutation in particular, such as clinical significance of P209L, before introducing findings from experiments?

- We added in description of the symptoms for patients with P209L mutation.

* "Using lentiviral transduction, we stably expressed wild-type human IFT139 or the P209L mutant in wild-type (WT) or knock-out (KO) cells" Murine cells were used to conduct transduction experiment, is there a reason why human KO cells were not used? How conserved are murine and human IFT139?

- Immortalized human cells are not Hedgehog signaling responsive.

- Mouse IFT139 is 85% similar to human IFT139 at the amino acid level (calculated using sequence information on UCSC & BLAST), and P209 is conserved between the two species.

* "To further quantify the structural defects observed in IFT139KO cells, we compared torsion (axoneme twist)" For these morphological analysis, higher resolution imaging may be useful, whether it is better resolved IF images or SEM. How were torsion and curvature measured, would it be possible to give further details in addition to stating Matlab scripts used.

- We added in the equation used to generate cilia torsion and curvature in the main text and in the figure legends, and added in more details in material and methods.

Lastly, can the authors speculate whether the peripheral IFT-A vs core IFT-A play distinct roles in controlling ciliogenesis and Hh signaling?

- We have some preliminary data that suggest there is a difference between individual IFTA proteins (IFT139 vs. IFT122 vs. IFT144) and their roles in controlling Hh signaling, the difference appears to be at the Gli-SuFu level. We hope we can continue with our cell-based research and work on that further.

Second decision letter

MS ID#: bio.062040

MS Title: IFT139 regulates Hedgehog signaling and cilia structure through ciliary protein localization.

Authors: Yulu Cherry Liu; Khatija Nishat; Zachary Klug; Jannatul Faimma Mia; Sara M Stump

Dear Dr Liu,

I have now reached a decision on the above manuscript.

The reviewer reports are shown at the bottom of this email or can be accessed, together with a copy of this decision letter, by going to:

As you will see, the reviewers raised a number of substantial criticisms that prevent me from accepting the paper at this stage.

Reviewer 1

Comments for the author

Review of MS ID#: bio.062040

IFT139 regulates Hedgehog signaling and cilia structure through ciliary protein localization.
Nishat et al

In this revised manuscript, the authors generally addressed reviewers' concerns, resulting in an improved report. There remains a major concern regarding proper statistical testing of results. Assuming use of correct statistical tests does not change results then the data support the conclusions drawn but the authors must justify the tests used- or reanalyze using proper statistical tests. There are a few minor concerns: the legend for figure 3D is missing from the text, the markers to indicate statistical significance are very small, and there are some grammatical errors that make the readability awkward in places.

Quality: Each figure contains proper controls and the experiments are conducted using appropriate methods. The data provided support the conclusions, assuming the use of correct statistical tests doesn't change the significance. In the Methods section, the authors state that they used student's t test to assess statistical significance of their results. However, there are several experiments where ANOVA (or equivalent) tests would be appropriate: comparison of more than two genotypes and/or treatments, repeated measures, or time course data.

Reproducibility: The inclusion of individual data points showing the technical reproducibility and data variability for the experiments is a great addition to this manuscript. The Methods section provides sufficient detail to permit evaluation and reproducibility.

Completeness and Scholarship: The authors' conclusions are supported by the presented data and the citations supporting and arguing against their conclusions have been properly referenced.

In this revised manuscript, the authors generally addressed reviewers' concerns, resulting in an improved report. There remains a major concern regarding proper statistical testing of results. Assuming use of correct statistical tests does not change results then the data support the conclusions drawn but the authors must justify the tests used- or reanalyze using proper statistical tests. There are a few minor concerns: the legend for figure 3D is missing from the text, the markers to indicate statistical significance are very small, and there are some grammatical errors that make the readability awkward in places.

Second revision

Author response to reviewers' comments

1. Statistical tests:

- Paired t-test used for: figure 2B, 2C, 2D, 2E, 2F, 2G, S1A and S3B. Justification: Paired t-test was used to compare a single variable between two groups. These experiments only have two groups.
- One-way ANOVA used for: figure 1B, 1D, 1E, 1G, 1H, 3A, 3B, 3C, 3D, S1C, S1D, S1E, S1F, S2B, S2C, S2D, S3C, and S3D. These experiments have more than two groups.
- All experiments retained the same statistical conclusion as before except for figure S1C.
 - Figure S1C: result was not statistically significant by one-way ANOVA. We addressed this in the manuscript and made sure not to draw any conclusion regarding Ptch2. This does not change the conclusion of the manuscript because Ptch2 is a paralog of Ptch1 with redundant function. Ptch1 is the main, established, receptor for Hedgehog pathway, and our data with Ptch1 is statistically significant with one-way ANOVA (Figure 1B).

2. Missing legend for figure 3D

- We double checked and figure 3D is not missing anything in the figure or in the text.

3. Small statistical significance markers

- We corrected this.
- For Figure 3A, 3B, and 3C we added in dashed lines to help better indicate statistical significances.

4. Grammar

- We double checked the manuscript for errors. We apologize if we missed any.

Third decision letter

MS ID#: bio.062040R1

MS Title: IFT139 regulates Hedgehog signaling and cilia structure through ciliary protein localization.

Authors: Yulu Cherry Liu; Khatija Nishat; Zachary Klug; Jannatul Faimma Mia; Sara M Stump

Dear Dr Liu,

I am happy to tell you that your manuscript has been accepted for publication in Biology Open, pending our standard publication integrity checks. It was accepted on 1st September 2025.

To see the reviewer's report and a copy of this decision letter, please go to: View Reviewer Comments

Your manuscript is now with our production department, and if we require anything further from you in terms of source files, we will be in touch shortly. Otherwise, you will receive proofs in due course.